# EMG-Based Estimation of Lower Limb Joint Angles and Moments Using Long Short-Term Memory Network

**DOI:** 10.3390/s23063331

**Published:** 2023-03-22

**Authors:** Minh Tat Nhat Truong, Amged Elsheikh Abdelgadir Ali, Dai Owaki, Mitsuhiro Hayashibe

**Affiliations:** Neuro-Robotics Lab, Department of Robotics, Graduate School of Engineering, Tohoku University, Sendai 980-8579, Japan

**Keywords:** electromyography, recurrent neural network, biomechanics, joint moment estimation, joint angle estimation

## Abstract

One of the fundamental limitations in human biomechanics is that we cannot directly obtain joint moments during natural movements without affecting the motion. However, estimating these values is feasible with inverse dynamics computation by employing external force plates, which can cover only a small area of the plate. This work investigated the Long Short-Term Memory (LSTM) network for the kinetics and kinematics prediction of human lower limbs when performing different activities without using force plates after the learning. We measured surface electromyography (sEMG) signals from 14 lower extremities muscles to generate a 112-dimensional input vector from three sets of features: root mean square, mean absolute value, and sixth-order autoregressive model coefficient parameters for each muscle in the LSTM network. With the recorded experimental data from the motion capture system and the force plates, human motions were reconstructed in a biomechanical simulation created using OpenSim v4.1, from which the joint kinematics and kinetics from left and right knees and ankles were retrieved to serve as output for training the LSTM. The estimation results using the LSTM model deviated from labels with average *R*^2^ scores (knee angle: 97.25%, knee moment: 94.9%, ankle angle: 91.44%, and ankle moment: 85.44%). These results demonstrate the feasibility of the joint angle and moment estimation based solely on sEMG signals for multiple daily activities without requiring force plates and a motion capture system once the LSTM model is trained.

## 1. Introduction

In robotics, accurate prediction of joint orientation and moments is essential. In developing an energy-efficient exoskeleton for persons with disabilities, joint moment estimation is crucial for physical interaction with users. Precise prediction of joint positions and moments allows the device to provide sufficient support for movements without interfering with users’ natural postures, ensuring the device’s safe operation, thus making it an appealing choice for assisting daily activities. Regarding bionic limb prostheses, although various sophisticated industrial products have been invented [1,2], the prosthetic control algorithm requires further development. Prostheses are supposed to detect and respond to users’ intentions, requiring high-performance orientation and moment control techniques to improve amputees’ quality of life. Hence, accurate prediction of joint angles and moments that align with users’ motor control strategies would significantly enhance the inputs to such control schemes.

Electromyography (EMG), which can be easily measured with commercially available wearable sensors, reflects muscle activity and thus can correlate to joint moments [3], making EMG an ideal information source for analyzing human movements. Additionally, following remarkable developments in recent years, deep learning algorithms have enabled robust and accurate estimations from measurable signals by training a black box to map the inputs to the outputs [4,5,6,7]. Previous studies employing machine learning techniques in EMG-driven applications have proven their accomplishments in making predictions, though most of them primarily focus on human kinematics estimation [8,9,10]. Among those techniques, one type of neural network called a recurrent neural network (RNN) [11,12,13] is capable of time-dependent processing signals and making predictions based on past information. However, traditional RNNs may not be able to handle the patterns of human movements that have long-term dependencies. Meanwhile, a Long-Short-Term Memory (LSTM) network [14] is a type of RNN employing a memory cell that can retain information for long periods, making it better equipped to handle the complexities of human movements.

We aim to evaluate a simple single LSTM model using features extracted from surface EMG (sEMG) signals to estimate human joint angles and moments during different daily living activities. This paper is organized as follows, Section 2 shows some research related to this work. In Section 3, the techniques utilized for acquiring the experimental data will be addressed, Section 4 shows and discuss the results of the study, and finally, Section 5 is our closing remarks.

## 2. Related Works

sEMG signals are used to decode joint kinematics and kinetics owing to their noninvasive measurement and broad availability. Several attempts have been made to map sEMG signals to joint angles or torques [15,16,17,18]. There is also an attempt to map sEMG signals to the ground reaction force [19]. In [15], Staudenman et al. proposed two configurations of high-density EMG (HD-EMG) electrodes (monopolar and bipolar sets) for muscle force estimation. Their study involved isometric arm extension movement in which an electrode grid was attached to the middle of the upper arm, above the triceps brachii muscle. The results of this study suggested that the EMG amplitudes recorded from the activity of the triceps brachii muscle can be compared directly with the elbow’s extension moment.

Sartori et al. introduced a model-based framework, namely the EMG-driven musculoskeletal model [20,21], to estimate the joint moment and joint stiffness consistent with experimental data. In real-time, the model computed the forces in 13 muscle-tendon units and displayed the resulting moments about 3 joint DoFs. This technique can cover a variety of movements as long as the parameters for the model are identified. However, because the joint moments are the sum of individual muscular components, the activity of all the muscles contributing to joint movements should be considered, which often requires a large set of EMG electrodes.

Liu et al. [22] extracted mean absolute value (MAV), root means square (RMS), waveform length (WL), and energy percentage (EP) from six muscles’ sEMG signals and used them to compare feed-forward neural network (FNN) and convolutional neural network (CNN) for knee moment estimation during gait. They also tested the CNN model using the sEMG without extracting any features. Their feature-based CNN model had the highest performance compared to the other two models, with root mean square error (RMSE) 5.88 ± 0.22 and *R*^2^ of 0.978 ± 0.002. They also found out that using EMG features reduced training time significantly.

In [23], Dongwon Kim et al. employed a model-free approach to predict the moment and angle based on sEMG signals of a paired flexor and extensor contributing to wrist movement. They used an LSTM network to perform estimation by feeding the EMG features through multiple layers to obtain the angle and moment. The results from their study show a favorable agreement in moment estimation (95%) and angle prediction (85%), encouraging the use of the LSTM network to decode natural human movements. However, this study focused only on a single joint. Thus, we were motivated to investigate a similar system applied to multi-joint inverse dynamics for the lower limbs.

There is a recent trend of combining EMG-driven muscle modeling and neural networks. The hybrid model used a CNN to map sEMG to specific muscle activation, which was used together with neuromusculoskeletal model components to compute knee torque [24]. A tradeoff exists between muscle model dependency and data-driven neural networks [25]. In [26], LSTM models were trained with muscle electromyography signals and lower limb joint angles. Hip flexion/extension, hip abduction/adduction, knee flexion/extension, and ankle dorsiflexion/plantarflexion torques were predicted, including transfer learning capability. This paper shows interesting results and is useful for some applications. However, it requires the joint angles information from the motion capture system as neural network input, and it limits the portability and mobility of the system. In contrast, in our proposition, we attempt to keep the portable EMG sensing only without needing a non-portable measurement system.

## 3. Methodology

Figure 1 illustrates how data were processed to train the LSTM network model. Several pre-processing steps were conducted to extract EMG features, joint moments, and angles.

### 3.1. Data Acquisition Process

Three acquisition systems were used in this study. First, the human translations were traced using the Optitrack system, an optical motion capture system that tracks the three-dimensional trajectories of reflective markers mounted on the subject at 100 Hz. GRFs were measured using two multi-axis AMTI force plates, each with a sampling frequency of 100 Hz. The force plate data were filtered with a fourth-order Butterworth filter with a cutoff frequency of 6 Hz. Meanwhile, a 16-channel Delsys Trigno wireless sEMG acquisition system records the sEMG signals from 14 muscles of interest at 1.1 kHz.

### 3.2. Experiment

The experiment comprises three stages: pre-experiment, static measurements, and dynamic measurements. The experiment was conducted with six healthy males with different anthropometric properties (height: 176 ± 9.2 cm and weight: 72.8 ± 11.7 kg); all subjects were informed about the procedure before the experiment started. The ethical committee of Tohoku University approved the experimental protocol under 22A-3.

Before the experiment, subjects were asked to wear tight T-shirts and swim trunks to reduce marker errors while recording their motions. A total of 14 Delsys Trigno sEMG sensors were placed on the subjects’ lower extremities. Muscle activities on both legs were investigated. Therefore, for each leg, the sEMG signals from seven muscles of interest were recorded, namely, the rectus femoris (RF), biceps femoris (BF), vastus lateralis (VL), semitendinosus (ST), tibialis anterior (TA), gastrocnemius medialis (GAS), and soleus (SOL), as shown in Figure 2. SENIAM recommendations [27] were followed for sensor placements to obtain the best signals.

After installing the sEMG sensors, 39 markers were attached to each subject at positions specified following a marker set available in OpenSim. Then, the subjects were requested to hold a reference pose (T-pose) to provide data for a subject-specific model in the OpenSim simulation. Eight markers were used only for the static measurement: knee lateral, knee medial, ankle lateral, and ankle medial on both legs, and these were removed after the static measurements were completed, while the remaining markers were used for static and dynamic trials. Figure 3 shows the 31 markers used for the dynamic measurements.

In the dynamic measurement phase, the participants were instructed to step on two force plates and perform the following sequences: squatting ten times, picking up an object and putting it down ten times, then sitting on a chair and standing up ten times. The participants repeated each action five more times to provide more data. Finally, they performed the actions in a random order during the remaining time of the experiment. Regarding dynamic measurements, we recorded experimental data for approximately 5 min without stopping, and our volunteers completed the tasks at their own pace.

### 3.3. Data Processing

As a biomedical signal, EMG is inherently noisy and thus requires pre-processing as a mandatory step to make the signals reliable before extracting features. The sEMG signals corresponding to the muscles of interest were filtered in the following manner:Passing raw signals through a fourth-order bandpass filter with a frequency range from 20 to 450 Hz [28].Passing signals through a second-order infinite impulse response (IIR) notch digital filter with a cutoff frequency of 50 Hz [28].Differentiating the signals to make them more stationary than the original [29].Normalizing the signals concerning the maximum value to maintain the signal between 0 and 1 [30].

The success of the predictive system depends mainly on the chosen EMG features. Several features could be extracted from EMG pre-processed data, such as integrated EMG (IEMG), mean absolute value (MAV), waveform length (WL), simple square integral (SSI), root mean square (RMS), myopulse percentage rate (MYOP), autoregression model (AR) coefficients, and cepstrum coefficients (CC) [29]. According to [31], a combined feature set, including AR coefficients, RMS, and a simple time-domain (TD) statistic showed a consistent improvement in classification accuracy. Consequently, in this work, we employed EMG features consisting of sixth-order AR coefficients, RMS, and MAV, resulting in eight features for each sEMG channel. In total, 112 features (14 sensors, 8 features each) were calculated using a 100 ms sliding window with a 50 ms step. These features were calculated using the following equations:(1)$$x\left(t\right)=\sum_{p=1}^{P}\left(a_{p}x\left(t-p\right)+\epsilon \right)$$
(2)$$RMS=\sqrt{\frac{1}{N}\sum_{t=1}^{N}x{\left(t\right)}^{2}}$$
(3)$$MAV=\frac{1}{N}\sum_{t=1}^{N}\left|x\left(t\right)\right|$$
where x(t) is the sample value at time *t*, *P* is the order of the AR model (in this work, *P* is 6), $a_{p}$ is the *p*th order AR coefficient, ϵ is the white noise, and *N* is the number of samples.

### 3.4. OpenSim Simulation

OpenSim [32] is a freely available, user-extensible software system that allows users to develop models of musculoskeletal structures and create dynamic simulations of movement. Various biomechanical applications have been developed based on this open-source software [33,34]. In this study, the Scale Tool, Inverse Kinematics (IK) Tool, and Inverse Dynamics (ID) Tool were used to obtain the joint angles and torques.

First, by applying OpenSim’s Scale Tool, a generic musculoskeletal model (gait2392) [35] was scaled based on a comparison of experimental marker data with virtual markers to generate a subject-specific model. After that, the IK tool was used to get the joints angles by solving the following weighted least squares equation:(4)$$\min_{q}\left[\sum_{i\in markers}w_{i}{∥x_{i}^{exp}-x_{i}\left(q\right)∥}^{2}+\sum_{j\in unprescribed\;coords}\omega {\left(q_{j}^{exp}-q_{j}\right)}^{2}\right]$$
where *q* is the coordinate vector to be solved, $w_{i}$ is the marker *i* weight, $x_{i}^{exp}$ is the experimental marker *i* 3D position, $x_{i}\left(q\right)$ is the corresponding virtual marker 3D position, $\omega_{i}$ is the coordinate *i* weight, and $q_{j}^{exp}$ is the experimental coordinate value.

Finally, OpenSim’s ID tool was used to estimate each joint’s net force and moment during the movement using kinematics information and the GRFs applied to the model at the calcaneus bones to solve the following ID, Equation (Equation 5).
(5)$$M\left(q\right)\ddot{q}+C\left(q,\dot{q}\right)+G\left(q\right)=\tau$$
where *N* is the number of degrees of freedom, $q,\dot{q},\ddot{q}\in R^{N}$ are vectors of generalized positions, velocity, and acceleration, $M\left(q\right)\in R^{N\times N}$ is the system mass matrix, $C(q,\dot{q}$) $\in R^{N}$ is the vector of Coriolis and centrifugal forces, $G\left(q\right)\in R^{N}$ is the vector of gravitational forces, and $\tau \in R^{N}$ is the vector of generalized forces.

### 3.5. Dataset

EMG features were computed at 20 Hz, while joint angles and moments were recorded at 100 Hz. Joint data were downsampled to match the features recording frequency, then passed through a sixth-order Butterworth lowpass filter with a cutoff frequency of 6 Hz. Features and labels are time-dependent; we stacked them together to form a sequence dataset. The dataset is divided into a training set (first 60%) for model training, validation (the following 20%) to validate trained models and testing (the last 20%) for the final evaluation of the trained model. Lastly, features were scaled between 0 and 1 using the training set only, and joint angles were scaled to make the training faster.

### 3.6. Evaluation Metrics

The performance of the proposed LSTM network model was evaluated by examining three metrics:(1)Coefficient of determination (*R*^2^) to determine how well the model fits the variance. However, it does not account significantly for the offset.(2)Root mean square error (RMSE) is affected by the offset between the actual and estimated values. The main drawback of using RMSE is that comparing results between different subjects and joints will result in misleading information because subjects and joints will have distinct ranges depending on the subject’s anthropometrics and other factors, such as the subject’s patterns, to achieve the tasks.(3)Normalized root mean square error (NRMSE) to make results’ intrasubject and intersubject links more meaningful.

These metrics were computed using the following equations:(6)$$R^{2}=1-\frac{\sum_{t=1}^{N}{\left(\hat{y_{t}}-y_{t}\right)}^{2}}{\sum_{t=1}^{N}{\left(\bar{y}-y_{t}\right)}^{2}}$$
(7)$$RMSE=\sqrt{\frac{1}{N}\sum_{t=1}^{N}{\left(\hat{y_{t}}-y_{t}\right)}^{2}}$$
(8)$$NRMSE=\frac{RMSE}{max\left(y_{t}\right)-min\left(y_{t}\right)}$$
where $\hat{y_{t}}$ is the predicted value at time *t*, $y_{t}$ is the measured value at time *t*, and $\bar{y}$ is the mean value of the observation.

### 3.7. LSTM Model

The LSTM is an RNN model that takes past information and builds long and short-term memories to predict future outputs based on the given data. The model’s performance is affected by the LSTM network’s hyperparameters, such as the model’s architecture, batch size, validation set, dropout, and activation units. There is no specific rule for determining the best hyperparameter combination. Training a deep neural network is a repetitive task that requires adjusting the hyperparameters and retraining the model until a suitable combination is identified to yield desirable results. The model was tuned to achieve high performance with a small number of trainable parameters to avoid overfitting. The LSTM model in this study had two stacked LSTM layers; each contains eight hidden units and a 30% dropout probability. The output layer is dense with eight neurons; each label will have a specific neuron assigned to it. The time step is equivalent to 1 s (20-time steps) and estimates the labels at the following time step (50 ms ahead of time). The model is trained using a NADAM optimizer with a reduced learning rate starting from 0.003 and a reduced factor of 70%. The mean square error (MSE) loss function was used, and the batch size was only eight.

## 4. Results and Discussion

### 4.1. Experimental Motion Analysis

Figure 4 shows the signal distribution of data per task and side for each subject. These data are collected during the first ten consecutive actions of each task. All subjects performed similar tasks, however, there are some intersubject differences in joint usage and its combination for completing the same task. Even in this situation, the joint angle and moment estimation performance is evaluated for different tasks.

#### 4.1.1. Squat

All subjects have comprehensive knee flexion when squatting; S1 and S3 applied higher knee flexion when squatting, yet S1’s normalized knee moment is more significant than S3’s because each subject combines various joints to complete a particular action, including the upper body to control the center of mass and prevent falling; this phenomenon is called synergy [36]. S1 flexed his knees to approximately 140°, greater than the other subjects’ knees extension, which resulted in the greatest knee extension moment exceeding 1.5 times the subject’s weight. Except for S2’s left knee moment, all other knee labels grow greater than the subjects’ weight. S3 tends to extend his knees, resulting in a 0.5 Nm/kg flexion moment on the left knee. Even though his knees have similar knee flexion profiles, the right knee extension moment grows close to 1.5 Nm/kg. In contrast, his left knee extension moment was only around 1 Nm/kg. He also extended his knees after each action, resulting in knee extension angles and moments, especially for the left side.

Ankle actions were more asymmetric than the knees. S1 had the greatest ankle plantar flexion moment with around 1 and 0.8 Nm/kg for both left and right ankles. The outlier values are represented when a subject performs an action one or two times more intensely than the other. S2’s ankles had the lowest plantar flexion moment, with less than 0.4 Nm/kg, with asymmetric dorsiflexion angles.

#### 4.1.2. Pick Up an Object

Unlike normal walking, where the patterns and range of motions are relatively comparable for both intersubject and intrasubject cases, for a task such as picking an object from the ground, some subjects applied unique strategies, depending on many factors such as the subject’s anthropometrics, the position of the object and subject’s body orientation concerning the object. S1, S4, and S6 have similar knee actions pattern for this task; they lower their bodies by flexing their knees (approximately −20° to −120°), similar to when squatting but with less knee extension moment compared to when squatting but with greater ankle dorsiflexion moment. Those subjects’ ankles only exercised dorsiflexion orientation. Still, unlike S4 and S6, S1 has a smaller ankle angle span. Nevertheless, he applied a high ankle plantar moment, especially on his right ankle, exceeding his body mass.

S2 and S5 slightly flex their knees to pick up the object (approximately 0° to −30°) and rely on lowering their upper body, which results in a knee flexion moment for most of the action’s time. Both subjects had intrasubject asymmetric ankles angles and moments, especially S2, whose left ankle was in the plantar-flex state for most of the action’s time, reaching −10°. S3 flexed his knees close to −60° when picking the object and extended them after it. This subject’s left knee only had an extension moment, unlike the right knee. The ankle motion of S3 was similar to S2 with asymmetric ankle angles. His right ankle was always in a dorsiflexion position and had a small range of motion compared to the right ankle, with a greater ankle plantar flexion moment.

#### 4.1.3. Sit Stand

Like the squat, subjects flex their knees in a comprehensive range when sitting on chairs but with small flexion moments. The lower knee angles’ and the zero knee moment medians are because subjects tend to sit for longer than the standing time. Like the previous two actions, S1’s left ankle plantar flexion moment is more significant than the subject’s body mass and greater than the right ankle moment, which means that this subject relies heavily on his left ankle to support the action compared to other subjects.

### 4.2. Estimation Performance with Personalized Intrasubject Models

Table 1, Table 2 and Table 3 illustrate the personalized intrasubject models’ on the test sets. The last row shows the mean ± standard deviation (Std) of the left and right labels. The *R*^2^ and NRMSE tables are used for intrasubject comparison. At the same time, the RMSE is suitable for reading each label’s actual error value.

The models estimated knee angles with high accuracy and low variance, achieving 97.25% ± 1.46% *R*^2^ and 4.95% ± 1.1% NRMSE, which is the best performance among other labels. The maximum difference between left and right knee angles is 0.52% *R*^2^ and 0.30% NRMSE in S2. Most of the variance came from the intersubject compassion. S1 achieved the highest results, with 98.48% *R*^2^, 3.2% NRMSE, and 3.73° RMSE on the right side, with a slight difference from the left side. In contrast, other models achieved remarkable knee angle estimation results above 97.4% *R*^2^ and NRMSE less than or equal to 5.6%, except for S5, where the model achieved lower performance than other subjects. Compared to the knee angle estimation, knee moment estimation’s *R*^2^ had dropped to 94.9% ± 1.4%. However, the NRMSE improved to 4.83% ± 0.88%. S3’s right knee moment estimation achieved the best results, with 97.15% *R*^2^ and 3.7% NRMSE, but his right knee had an RMSE of 0.076 Nm/kg, which is higher than the left knee moment by 0.002 Nm/kg.

Regarding the ankle labels, the intersubject and intrasubject variance appears more than knee labels with lower performance. S3’s ankle angles had the highest *R*^2^, with approximately 95.5% for both sides. However, they achieved 6.3% and 6.2% NRMSE for both left and right ankle angles, which is greater than S1’s and S2’s left ankle angles. The model delivered similar results for S3, although the subject tends to have greater left ankle dorsiflexion than the right ankle. A 3.67% *R*^2^ and 1.7% NRMSE difference between S2’s ankles angles is the most enormous intrasubject variance for the ankle labels.

Ankle moment has the worst estimation results, with 85.44% ± 8.14% *R*^2^ and 8.46% ± 1.98% NRMSE. S1’s right ankle moment had the highest ankle moment estimation performance with 96.19% and 5.1% *R*^2^ and NRMSE. Although the model could estimate S4’s left ankle angles with 90.45% *R*^2^ and 8.3% NRMSE, S4’s left ankle moment estimation had the worst estimations, with 69.04% *R*^2^ and 12.2% NRMSE. S4’s ankle moment estimation also had the highest intrasubject effect between left and right ankles. S2’s model could estimate the right ankle moment better than the left ankle moment. Only S1’s and S5’s ankle moments estimations were better than their ankle angle estimation for both *R*^2^ and NRMSE metrics.

Figure 5 and Figure 6 show the measurement against the estimation for S1 and S2 labels. S1 applied a high and relatively smooth ankle moment similar to the knee angles; therefore, the model could estimate the ankle angles accurately, unlike S2, which had issues maintaining a smooth ankle moment line. The models could highly estimate motion with an extensive range better than the ones with quick changes in direction, which is why the knee angle estimation had the highest estimation results for all models. S2’s model failed to estimate ankle moments correctly. However, it followed the moment’s trend, resulting in the misleading right ankle moment *R*^2^ score. Moreover, it made a mistake in estimating the knee angle when the subject started sitting on the chair but could find the correct angle value before the subject started standing from the chair.

## 5. Conclusions

The standard approach in biomechanics for estimating lower limb joint moments is applying inverse dynamics based on motion capture and force plate information. However, the force plate’s ground reaction force is available only for a small area of the force plate. This work investigated the LSTM network for the kinetics and kinematics prediction of human lower limbs when performing different activities without using force plates. After the learning, the neural network only requires the measured sEMG signals for accessing the estimated joint moment and angles of the lower limb. We have proposed this new framework for lower limb joint moment estimation and verified its feasibility through experiments. We measured surface electromyography signals from 14 lower extremity muscles to generate a 112-dimensional input vector from 3 sets of features: root mean square, mean absolute value, and sixth-order autoregressive model coefficient parameters for each muscle for the LSTM network to estimate knee and ankle joint angles and moments on both sides for eight participants.

Different subjects apply different knee and ankle strategies for the tasks, resulting in diverse datasets. Not all our subjects are athletes, resulting in an imbalance between left and right labels. The trained LSTM models are nonlinear and can estimate knee and ankle angles and moments. Despite these challenges, the LSTM models could estimate knee angles and moments with 97.25% and 94.9% mean *R*^2^ and 4.95% and 4.83% mean NRMSE. The models achieved results on ankle angle estimation of 91.44% mean *R*^2^ and 7.12% mean NRMSE. Out of 12 ankle moment estimations, only 4 outputs had more than 90% *R*^2^. The models achieved results on ankle moment estimation of 85.44% mean *R*^2^ and 8.46% mean NRMSE.

LSTM models demonstrated some bias, especially for ankle moments, compared to knee moments. Looking at the measured labels against the estimated labels in Figure 5 and Figure 6, we believe that the models find it difficult to estimate ankle moments because they find it hard to maintain their balance, which results in small changes in the moment, which was not the case for S1, who maintained smooth ankle moment curve.

Generally speaking, an LSTM model with 4488 parameters was trained for each subject. It could estimate eight different joint angles and moments for multitasking activities using only EMG features. This study has some limitations and drawbacks that need to be addressed in future work:We assumed that all the output measurements from the motion capture system, force plates, and OpenSim calculations were accurate and reliable.sEMG signals are susceptible to external factors such as sweat or motion artifacts, which are inherent issues when opting for the surface type. These factors may affect the quality of the signal.The number of movements is still limited. This system does not cover walking, jumping, or running activities, which are usually in high demand.The number of investigated joints is also insufficient. The hip joint was excluded from this study because the sEMG signals of the muscles of interest contributing to hip joints are challenging to access stably for the measurement.Results are not always consistent from one subject to another (intersubject variability).

## Figures and Tables

**Figure 1 sensors-23-03331-f001:**
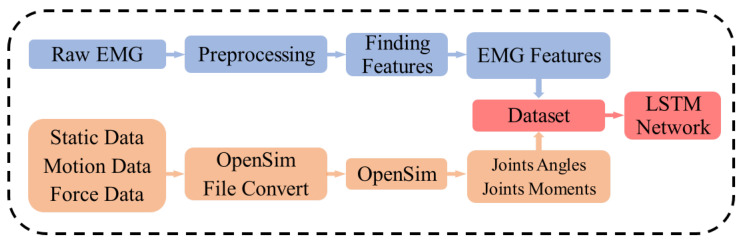
Data flow for training the LSTM network model.

**Figure 2 sensors-23-03331-f002:**
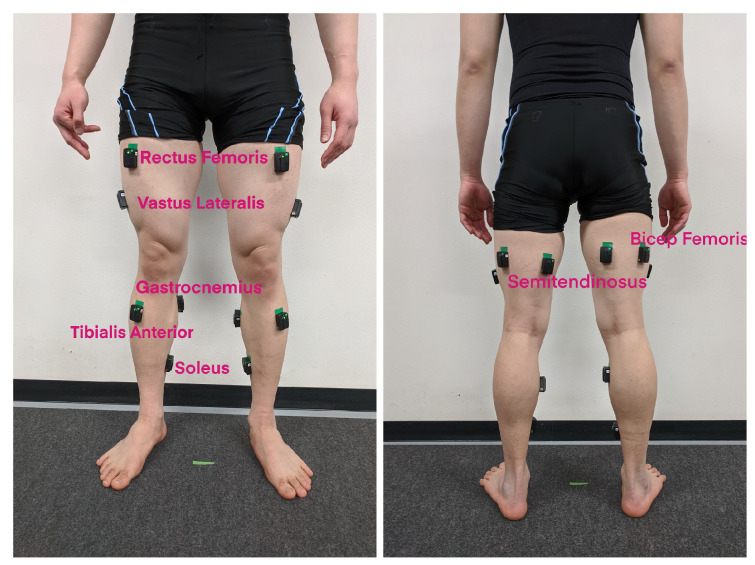
Fourteen sEMG sensors placed on the muscles of interest.

**Figure 3 sensors-23-03331-f003:**
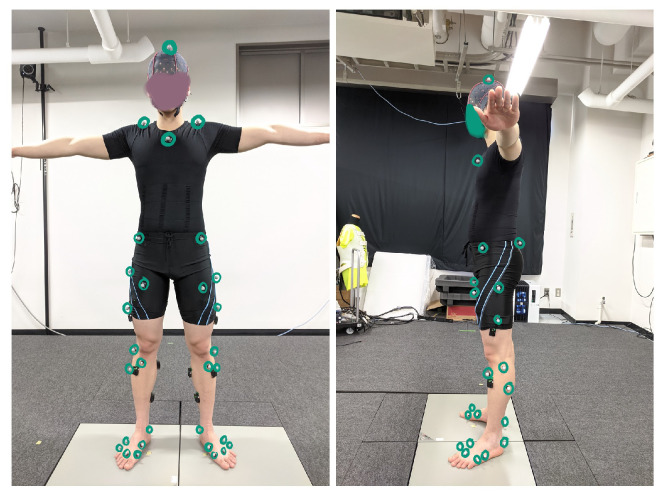
Mounting placement of markers on the subject for dynamic measurements.

**Figure 4 sensors-23-03331-f004:**
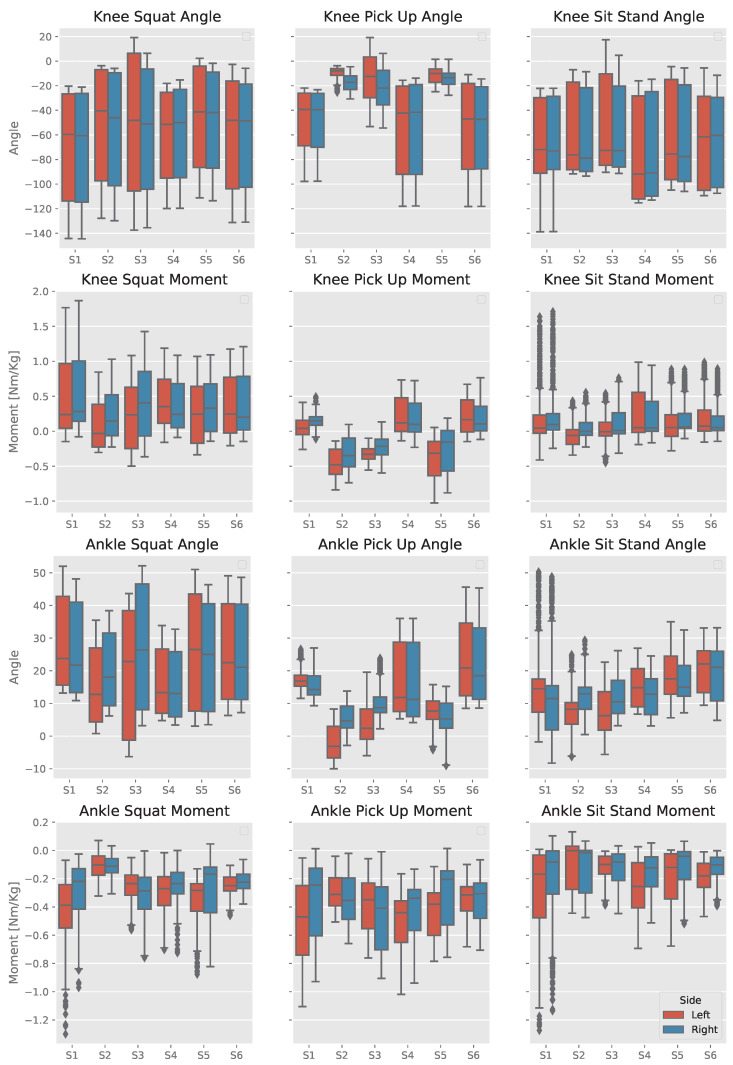
Distribution of the kinematics and kinetics data when each subject performed each task for the first ten times. Red boxes indicate the left side, and blue boxes indicate the right side.

**Figure 5 sensors-23-03331-f005:**
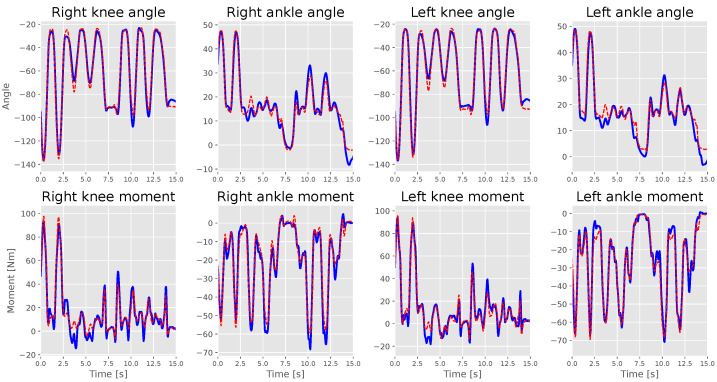
S1’s test set measured (straight blue lines) against estimations (dotted red lines) during an interval of 15 s.

**Figure 6 sensors-23-03331-f006:**
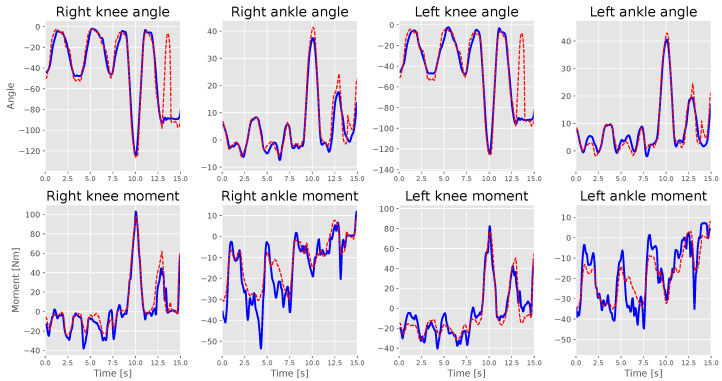
S2’s test set measured (straight blue lines) against estimations (dotted red lines) during an interval of 15 s.

**Table 1 sensors-23-03331-t001:** *R*^2^ score of the trained subjects’ personalized models.

	Angle				Moment			
	Knee		Ankle		Knee		Ankle	
	Left	Right	Left	Right	Left	Right	Left	Right
S1	98.40%	98.48%	94.61%	94.12%	94.33%	93.46%	95.27%	96.19%
S2	97.52%	97.00%	94.03%	90.36%	96.19%	96.09%	82.52%	90.79%
S3	97.80%	97.91%	95.58%	95.54%	96.64%	97.15%	87.74%	89.22%
S4	97.48%	97.73%	90.45%	89.81%	94.39%	95.60%	69.04%	78.74%
S5	94.19%	94.29%	85.33%	85.75%	94.73%	93.38%	88.69%	90.55%
S6	98.21%	97.99%	90.99%	90.65%	94.05%	92.83%	80.19%	76.39%
Mean ± Std.	97.25% ± 1.46%		91.44% ± 3.48%		94.9% ± 1.4%		85.44% ± 8.14%	

**Table 2 sensors-23-03331-t002:** NRMSE score of the trained subjects’ personalized models.

	Angle				Moment			
	Knee		Ankle		Knee		Ankle	
	Left	Right	Left	Right	Left	Right	Left	Right
S1	3.20%	3.20%	4.20%	4.40%	4.10%	4.60%	5.70%	5.10%
S2	4.90%	5.20%	5.90%	7.60%	4.30%	4.50%	10.20%	7.70%
S3	4.30%	4.10%	6.30%	6.20%	4.40%	3.70%	8.20%	8.20%
S4	5.60%	5.40%	8.30%	8.70%	5.40%	4.70%	12.20%	10.00%
S5	6.60%	6.50%	8.40%	8.80%	4.50%	4.70%	8.00%	8.20%
S6	5.10%	5.30%	8.40%	8.20%	6.40%	6.60%	7.70%	10.30%
Mean ± Std.	4.95% ± 1.1%		7.12% ± 1.66%		4.83% ± 0.88%		8.46% ± 1.98%	

**Table 3 sensors-23-03331-t003:** RMSE of the trained subjects’ personalized models.

	Angle				Moment [Nm/Kg]			
	Knee		Ankle		Knee		Ankle	
	Left	Right	Left	Right	Left	Right	Left	Right
S1	3.76	3.73	2.23	2.49	0.073	0.079	0.065	0.058
S2	6.00	6.21	2.80	3.05	0.064	0.066	0.057	0.053
S3	6.73	5.78	3.12	3.08	0.074	0.076	0.060	0.073
S4	5.75	5.66	2.70	2.72	0.075	0.063	0.132	0.096
S5	8.35	8.25	4.88	4.66	0.102	0.104	0.073	0.069
S6	5.96	5.90	3.19	3.21	0.080	0.080	0.048	0.062
Mean ± Std.	6.01 ± 1.40		3.18 ± 0.80		0.078 ± 0.013		0.071 ± 0.023	

## Data Availability

Raw data for subject 1 are accessed on 29 November 2022 and available at https://github.com/Amged-Elsheikh/EMG-based-Estimation-of-Lower-Limb-Joint-Angles-and-Moments-Using-Long-Short-Term-Memory-Network along with the necessary codes and briefly supplementary video for the paper.

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
