# Peer review of "EMG-Based Estimation of Lower Limb Joint Angles and Moments Using Long Short-Term Memory Network"

_sensors, 2023, doi:10.3390/s23063331_

Round 1

Reviewer 1 Report

The article discusses a study that investigated the use of Long Short-Term Memory (LSTM) networks to predict joint angles and moments in human lower limbs during different activities using surface electromyography (sEMG) signals. Joint moments cannot be directly obtained without affecting motion, but can be estimated using external force plates. However, the study found that the LSTM model could accurately estimate joint angles and moments based solely on sEMG signals, without requiring force plates or a motion capture system, with high R2 scores for knee and ankle angles and moments. The paper is well presented. I would be interested in seeing a higher number of subjects or at least a justification for just 6 males.

More details:

1. The study found that LSTM model could accurately estimate joint angles and moments based solely on sEMG signals, without requiring force plates or a motion capture system.
2. The topic is indeed original and relevant to researchers trying to estimate kinetics using EMG. For example movement intention.
could be collected using this method for robotic exoskeletons.
3. The paper adds to the extensive research being done on machine learning based classification of biosignals.
4. The methodology is fine. I realise the authors were testing the algorithm as a pilot study with 6 subjects. However, I would indeed be interested in seeing the model tackle data from a larger population with a lot of variability in EMG.
5. the conclusions consistent with the evidence and arguments presented and they address the main question posed.
6. the references are appropriate.
7. The figures are appropriate.

Author Response

Comment

I would be interested in seeing a higher number of subjects or at least a justification for just 6 males.

Reply

First, thank you for your many positive comments on our paper, as written in your review report. We answer the one concern you raised regarding the subject number below.

One of the biggest challenges in the study is data collecting, cleaning, and processing. While 6 subjects are not enough to train an intersubject generic model, it was enough to show the stable individual subject estimation capabilities and limitations of an LSTM model trained on sEMG features alone. For studies with repetitive motion, such as gait analysis, 6~8 subjects might be enough, as was the case for previous research papers, especially for new engineering method propositions. However, for studies with different exercises where each subject will have different strategies, the number of required subjects can overgrow to create a general model.

Having only male subjects is one of the study's limitations. However, we could not have female participants due to the required physical contact with the subject when installing the motion capture markers and the sEMG sensors.

This paper is the first proposition paper on this approach, for verifying the feasibility of the proposed method, please let us report on this first result. And more detailed analysis and more transferable estimation over inter-subject estimation capabilities should be further studied in future research.

Reviewer 2 Report

In this paper a Long Short-Term Memory (LSTM) network for the kinetics and kinematics prediction of human lower limbs when performing different activities without using force plates after the learning. However the current work of this manuscript is not comprehensive enough to be accepted for its publication in the journal, and some major improvements is necessary in the content of the different sections.

1.       In section 1, the necessity and innovations of the paper were not described clearly. The previous studies about the application of LSTM for the estimation of EMG should be discussed. The latest references about this studies should be cited.

2.       Following the above points, another major question will be what the existing methods are and

how the proposed method could compare with one or two representative ones from the literature?

Author Response

Comment: 

  1. In section 1, the necessity and innovations of the paper were not described clearly. The previous studies about the application of LSTM for the estimation of EMG should be discussed. The latest references about this studies should be cited.
  2. Following the above points, another major question will be what the existing methods are and how the proposed method could compare with one or two representative ones from the literature?

Reply

  1. Thank you for your feedback on our paper. The main point was to improve the introduction section (related works introduction), especially regarding previous studies about applying LSTM for estimating EMG (I guess it is for joint moment prediction using EMG). We have conducted additional searches for previous works for the latest references on this topic. We have improved the description regarding the related works introduction.  As highlighted in the manuscript, we have added four new recent references on the work of LSTM with EMG input. And the differences in approach are discussed.
  2. The above discussion on introducing previous studies about the application of LSTM for the estimation of the joint moment by using EMG. We have described the existing method's approach and discussed the differences. In comparison, all different methods employ different information for training neural networks, and it is difficult to compare with the same computational condition. In our proposition, we attempt to keep the portable EMG sensing only without needing a non-portable measurement system like a motion capture system because the capturable motion area is limited in a specific laboratory space.

Reviewer 3 Report

the paper is correct and well-founded.

Author Response

Thank you very much for your positive feedback about our research paper.

Reviewer 4 Report

1.     sEMG is used in lines 55 and 145 of the article, while EMG is used elsewhere. is there a difference between the two in this article?

2.     In line 96 of the article, the filter used is only given a cutoff frequency without specifying which filter it is.

3.     In lines 120-122 of the article, several actions were selected for the experiment, please explain the reasons for choosing these actions for the experiment.

4.     In line 145 of the article, it says that the 112 features are from 8 sensors with 14 features each, but from the description in the article, it seems that 14 sensors are used, and 8 features are extracted from each channel, is this correct?

5.     In Figure 1 of the article, torques are used, but in some parts of the text, moments are used. Do the two have the same meaning in the text, and if so, please use one of them consistently.

Author Response

Comments

  1. sEMG is used in lines 55 and 145 of the article, while EMG is used elsewhere. is there a difference between the two in this article?
  2. In line 96 of the article, the filter used is only given a cutoff frequency without specifying which filter it is.
  3. In lines 120-122 of the article, several actions were selected for the experiment, please explain the reasons for choosing these actions for the experiment.
  4. In line 145 of the article, it says that the 112 features are from 8 sensors with 14 features each, but from the description in the article, it seems that 14 sensors are used, and 8 features are extracted from each channel, is this correct?
  5. In Figure 1 of the article, torques are used, but in some parts of the text, moments are used. Do the two have the same meaning in the text, and if so, please use one of them consistently.

Reply

  1. Both sEMG and EMG refer to the same signal. When mentioning sEMG, it emphasizes the source of the signal and the signal itself, while when mentioning EMG, only the focus is turned to the signal only. We started the Related work chapter by stating that sEMG is our focus. Following the comment from the reviewer, we will change the following:
    1. “EMG” to “sEMG” in lines 73, 81, 120, 125, 141, 323, 351 and 357.
    2.  “sEMG” to “EMG” in line 347
  2. We used a “second-order IIR notch digital filter.” We will add more description to the “notch” word used.
  3. We chose those movements to record lots of data with consistency without segmentation, as both feet are always kept on the ground. Moreover, those movements can generate an extensive range of motions for the knee and ankle to capture the prediction ability of the LSTM. 
  4. Sorry for this typing mistake. We used 14 sensors and extracted 8 features from each. We will fix it. Thanks for noticing it.
  5. Both Torque and Moment have the same meaning. We will change the figure to use Moment.